# Tumor Necrosis Factor (TNF) blocking agents are associated with lower risk for Alzheimer's disease in patients with rheumatoid arthritis and psoriasis

**Mengshi Zhou**[1], **Rong Xu**[2]*, **David C. Kaelber**[1,3], **Mark E. Gurney**[4]*

**1** Department of Population and Quantitative Health Sciences, School of Medicine, Case Western Reserve University, Cleveland, OH, United States of America, **2** Center for Artificial Intelligence in Drug Discovery, School of Medicine, Case Western Reserve University, Cleveland, OH, United States of America, **3** Departments of Internal Medicine and Pediatrics and the Center for Clinical Informatics Research and Education, The MetroHealth System, Cleveland, OH, United States of America, **4** Tetra Therapeutics, Grand Rapids, MI, United States of America

* rxx@case.edu (R.X.); mark@tetratherapeutics.com (M.E.G.)

**Data Availability Statement:** All relevant data are within the paper and its Supporting Information files.

## Abstract

This large, retrospective case-control study of electronic health records from 56 million unique adult patients examined whether or not treatment with a Tumor Necrosis Factor (TNF) blocking agent is associated with lower risk for Alzheimer's disease (AD) in patients with rheumatoid arthritis (RA), psoriasis, and other inflammatory diseases which are mediated in part by TNF and for which a TNF blocker is an approved treatment. The analysis compared the diagnosis of AD as an outcome measure in patients receiving at least one prescription for a TNF blocking agent (etanercept, adalimumab, and infliximab) or for methotrexate. Adjusted odds ratios (AORs) were estimated using the Cochran-Mantel-Haenszel (CMH) method and presented with 95% confidence intervals (CIs) and p-values. RA was associated with a higher risk for AD (Adjusted Odds Ratio (AOR) = 2.06, 95% Confidence Interval: (2.02–2.10), $P$-value <0.0001) as did psoriasis (AOR = 1.37 (1.31–1.42), $P$ <0.0001), ankylosing spondylitis (AOR = 1.57 (1.39–1.77), $P$ <0.0001), inflammatory bowel disease (AOR = 2.46 (2.33–2.59), $P$ < 0.0001), ulcerative colitis (AOR = 1.82 (1.74–1.91), $P$ <0.0001), and Crohn's disease (AOR = 2.33 (2.22–2.43), $P$ <0.0001). The risk for AD in patients with RA was lower among patients treated with etanercept (AOR = 0.34 (0.25–0.47), $P$ <0.0001), adalimumab (AOR = 0.28 (0.19–0.39), $P$ < 0.0001), or infliximab (AOR = 0.52 (0.39–0.69), $P$ <0.0001). Methotrexate was also associated with a lower risk for AD (AOR = 0.64 (0.61–0.68), $P$ <0.0001), while lower risk was found in patients with a prescription history for both a TNF blocker and methotrexate. Etanercept and adalimumab also were associated with lower risk for AD in patients with psoriasis: AOR = 0.47 (0.30–0.73 and 0.41 (0.20–0.76), respectively. There was no effect of gender or race, while younger patients showed greater benefit from a TNF blocker than did older patients. This study identifies a subset of patients in whom systemic inflammation contributes to risk for AD through a pathological mechanism involving TNF and who therefore may benefit from treatment with a TNF blocking agent.

**Funding:** R.X. acknowledges support from Eunice Kennedy Shriver National Institute of Child Health & Human Development of the National Institutes of Health under the NIH Director's New Innovator Award number DP2HD084068, NIH National Institute of Aging R01 AG057557, R01 AG061388, R56 AG062272 and American Cancer Society Research Scholar Grant RSG-16-049-01 – MPC; Tetra Therapeutics provided support for the study in the form of salary for M.E.G. The specific roles of this author are articulated in the 'author contributions' section. M.E.G. acknowledges support from the National Institute of Mental Health awards MH091791 and MH107077. The funding sources had no role in the design and conduct of the study; collection, management, analysis, or interpretation of the data; preparation, review, or approval of the manuscript; or the decision to submit the manuscript for publication.

**Competing interests:** M.E.G is an employee of Tetra Therapeutics. This commercial affiliation does not alter the adherence of M.E.G to PLOS ONE policies on sharing data and materials.

## Introduction

Alzheimer's disease (AD) is the most common cause of dementia. Characteristic pathology in brain includes the presence of plaques (deposits of amyloid-β peptide) and tangles (intraneuronal deposits of hyperphosphorylated tau protein) [1, 2]. These occur in hippocampus and associational cortex, regions of the brain important for cognitive function. Accumulation of amyloid plaque occurs over many years and generally precedes accumulation of intraneuronal tangles and cognitive dysfunction [2]. Progression of disease is associated with neuronal degeneration, cortical thinning, and deepening and severe cognitive impairment. Neuroinflammatory changes around the amyloid plaques include reactive astrocytosis and microglial cell activation [3]. Microglia cells, the resident macrophages in the brain, derive embryonically from the same myeloid stem cells in bone marrow that give rise to monocytes and macrophages elsewhere in the body [4]. Macrophages are one of the important drivers of systemic inflammation through the overproduction of tumor necrosis factor (TNF) and other proinflammatory cytokines [5]. In the brain, elevation of TNF in cerebrospinal fluid collected from subjects with mild cognitive impairment is associated with progression to AD at 6 months follow up [6]. Multiple lines of evidence indicate that TNF may trigger or amplify aberrant microglia signaling in the brain [7–10] and thereby contribute to AD pathogenesis.

Multiple systemic inflammatory diseases are caused in part by the production of TNF by activated macrophages and these can be treated effectively with a TNF blocking agent. Approved indications for TNF blockers include rheumatoid arthritis (RA), ankylosing spondylitis, psoriasis, psoriatic arthritis, ulcerative colitis (UC) and Crohn's disease (CD) which are subtypes of inflammatory bowel disease (IBD) [5]. Inflammation in RA is localized to the synovial lining of the joints where synovial cell proliferation leads to a thickening of the lining, infiltration of activated macrophages and other inflammatory cells, elevated production of TNF, and irreversible destruction of the joint architecture and function [11]. TNF produced systemically may directly enter the brain through receptor-mediated transcytosis [12]. In mice, both types of TNF receptors, TNFR1 and TNFR2, participate in the transport of TNF across the blood brain barrier while other proteins present in blood are excluded. Thus, systemic production of TNF may directly affect inflammatory processes in the brain relevant to AD.

We therefore sought to test the hypothesis that systemic inflammation involving TNF is associated with increased risk for AD and that this could be mitigated by a TNF blocking agent. A previous case-control study of 8.5 million insured adults in the United States (US) reported increased risk for AD in patients with RA and that etanercept, a TNF blocking agent prescribed for the treatment of RA, was associated with reduction of risk for AD (Adjusted Odds Ratio (AOR) = 0.30; 95% Confidence Interval (CI) = 0.08–0.87. *P*-value = 0.02) [13]. The study was based on 9,253 patients with a diagnosis of AD in the private insurance database.

AD is underrepresented in US private insurance databases as most patients with AD are older than 65, an age in the US when the majority of Americans transition from employer provided, private insurance to Medicare, a national health insurance program. We therefore undertook a retrospective study using the IBM Watson Healthcare Explorys Cohort Discovery platform which contains electronic health records (EHR) from patients with private insurance, Medicare and Medicaid. The Explorys Cohort Discovery platform contains the de-identified EHR of nearly 56 million unique patients with age ≥ 18 years from 26 healthcare systems across all 50 states in the US from 1999 to 2018. We expanded our analysis to other inflammatory diseases and to additional drugs in the class of TNF blocking agents to assess their potential effect on risk for AD. Recent studies have shown that with this unique EHR database and built-in informatics tools, large hypothesis-driven case-control studies can be undertaken [14–

16]. Sufficient patient EHR were available for the evaluation of co-morbid AD in patients with RA or psoriasis that were treated with etanercept, adalimumab, or infliximab. Etanercept (Enbrel®) is a fusion of soluble TNF receptor 2 with the Fc portion of mouse immunoglobulin G1 [17]. Adalimumab (Humira®) is a fully human, anti-TNF monoclonal antibody [18], while infliximab (Remicade®) is a chimeric mouse-human monoclonal antibody in which the antigen combining region of a mouse anti-TNF monoclonal antibody is fused to a human Fc domain [19]. Such large, biologic drugs are not expected to have good access to brain after subcutaneous or intravenous injection due to exclusion of protein macromolecules by the blood brain barrier [12, 20].

## Methods

### Database description

We performed a retrospective case-control study using de-identified population-level EHR collected by the IBM Watson Health Explorys Cohort Discovery platform from 360 hospitals and 317,000 providers [15]. The de-identified data are accessed using the Explore application from the Explorys Cohort Discovery platform. Explorys collects data from multiple health information systems using a health data gateway (HGD) server. The data collected include patient demographics, disease diagnoses, medication history, findings, procedures, and laboratory test results. The EHR are de-identified in accordance with the Health Insurance Portability and Accountability Act (HIPAA) and the Health Information Technology for Economic and Clinical Health (HITECH) Act standards. After the de-identification process, the HGD server normalizes the data to facilitate search using the clinical ontologies from the United Medical Language System (UMLS) [21]. Specifically, systematized nomenclature of medicine–clinical terms (SNOMED-CT) are used to map disease diagnoses, clinical findings, procedures, and pharmacological drug classes [22]; individual drug names are mapped into RxNorm [23], and lab results and measurements are mapped into logical observation identifiers names and codes (LOINC) [24]. For HIPAA-compliant, statistical de-identification, the IBM Watson Health Explorys Cohort Discovery platform does not report cohort counts less than 10. At the time of the study, the Explorys Cohort Discovery platform contained over 64 million unique patients among which nearly 56 million were over 18 years of age (21% of the U.S. population).

### Study population

Patients were drawn from the Explorys platform and categorized based on their disease diagnoses and medication history. We defined patients as having AD if they had any encounter diagnosis in their contributing EHR of AD based on the SNOMED-CT disease name. A similar method was applied for identifying patients with the broader diagnosis of dementia. Since AD is a subdiagnosis of dementia within SNOMED-CT, the group of patients with dementia included all patients with a diagnosis of AD (**Table 1**). SNOMED-CT disease names also were used to identify patients diagnosed with one of eight inflammatory diseases: RA, ankylosing spondylitis, psoriasis, psoriatic arthritis, inflammatory bowel disease, ulcerative colitis, and Crohn's disease. Inflammatory bowel disease is a SNOMED-CT diagnostic category that includes all patients with the sub-diagnosis of ulcerative colitis but excludes patients with Crohn's disease. Patients diagnosed with more than one inflammatory disease were excluded from the analysis. For example, a patient diagnosed with RA who also was diagnosed with ankylosing spondylitis, psoriasis, psoriasis with arthropathy, inflammatory bowel disease, ulcerative colitis, or Crohn's disease would have been excluded from the analysis. The SNOMED-CT codes are listed in S17 Table. The effects of each inflammatory disease on AD or

**Table 1. Baseline characteristics of the study population.**

| Patient Characteristics | Study population | Alzheimer's disease | Dementia |
|---|---|---|---|
| **Total No. of patients** | 55,954,070 | 338,400 | 999,800 |
| **Gender (Number (%))** | | | |
| Female | 30,708,300 (54.88) | 216,530 (63.99) | 605,350 (60.55) |
| Male | 25,069,350 (44.80) | 121,770 (35.98) | 394,110 (39.42) |
| Unknown | 176,410 (0.32) | 100 (0.030) | 330 (0.030) |
| **Age (Number (%))** | | | |
| Adult (18 to 65) | 39,294,220 (70.23) | 13,300 (3.93) | 151,130 (15.12) |
| Senior (> 65) | 15,788,180 (28.22) | 308,930 (91.29) | 810,180 (81.03) |
| Unknown | 850,590 (1.52) | 16,120 (4.76) | 38,230 (3.82) |
| **Race (Number (%))** | | | |
| White | 32,209,260 (57.56) | 263,850 (77.94) | 773,810 (77.40) |
| Non-White | 23,744,810 (42.44) | 74,560 (22.03) | 225,990 (22.60) |
| **Insurance (Number (%))** | | | |
| Private | 19,854,540 (35.48) | 112,350 (33.20) | 351,420 (35.15) |
| Medicare | 7,047,380 (12.59) | 213,760 (63.17) | 578,670 (57.88) |
| Medicaid | 3,448,710 (6.16) | 21,030 (6.21) | 84,040 (8.41) |
| **Diagnoses (Number (%))** | | | |
| Rheumatoid arthritis | 514,440 (0.92) | 16,280 (4.81) | 47,270 (4.73) |
| Ankylosing spondylitis | 35,550 (0.060) | 460 (0.14) | 1,620 (0.16) |
| Psoriasis | 309,660 (0.55) | 3,340 (0.99) | 10,500 (1.05) |
| Psoriatic arthritis | 80,580 (0.14) | 650 (0.19) | 2,220 (0.22) |
| Inflammatory bowel disease | 279,040 (0.50) | 4,440 (1.31) | 13,770 (1.38) |
| Ulcerative colitis | 168,870 (0.30) | 2,650 (0.78) | 8,220 (0.82) |
| Crohn's disease | 201,870 (0.36) | 6,160 (1.82) | 12,270 (1.23) |
| **Drugs (Number (%))** | | | |
| Etanercept | 44,210 (0.080) | 250 (0.070) | 1,080 (0.11) |
| Adalimumab | 66,820 (0.12) | 320 (0.090) | 1,260 (0.13) |
| Infliximab | 40,290 (0.070) | 340 (0.10) | 1,100 (0.11) |
| Methotrexate | 216,200 (0.39) | 2,850 (0.84) | 9,550 (0.96) |

dementia were compared to a non-inflammatory disease group that comprised patients who had no diagnosis of any of the 8 inflammatory diseases listed above.

We further considered confounding environmental factors that are known to be associated with both TNF mediated inflammatory diseases and AD, including smoking, alcohol use, Body Mass Index (BMI), and obesity [25–32]. The large number of RA patients in the study population allowed us to validate the association of RA with AD after adjusting for the confounding factors of BMI/obesity (18.5–24.99 versus > = 30), smoking status (Non-smoker versus Tobacco user), and alcohol use status (Current drinker of alcohol versus Current non-drinker of alcohol), respectively.

A patient was considered "taking a medication" if at least one outpatient prescription for the medication had been written for the patient. Patients who took drug treatment were identified by searching the generic drug names. The Explorys Cohort Discovery platform has standardized all drug names to their generic names based on RxNorm. There are five TNF blocking agents that have been approved by US Food and Drug Administration (FDA): etanercept, adalimumab, infliximab, certolizumab pegol, and golimumab. Sufficient numbers of subjects (at least 10 patients in each cell in a two by two table for at least one strata) were available to study the effects of etanercept, adalimumab, and infliximab. For the analysis, subjects were

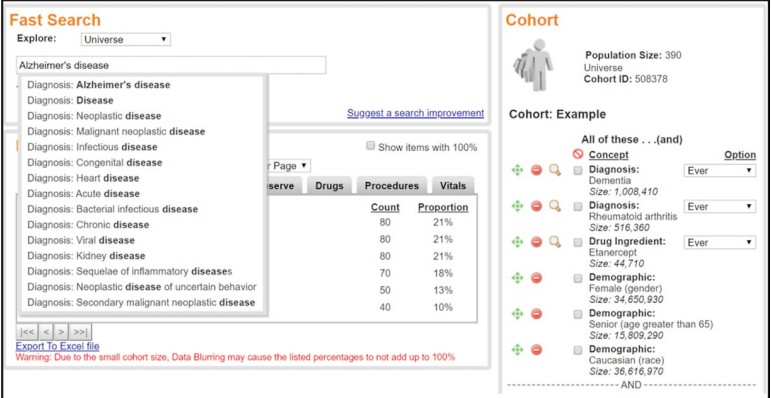

**Fig 1. A screen shot of cohort identification using the IBM Watson Health Explorys Cohort Discovery fast search tool.** This example selects female White patients with age > 65 years with a diagnosis code for dementia and a prescription for etanercept.

included if they were treated with a single TNF blocker but were excluded if they were treated with two or more TNF blocking drugs. For example, in the etanercept group, patients who also were prescribed adalimumab, infliximab, certolizumab pegol, or golimumab were excluded. Similarly, in the methotrexate group, patients who also were prescribed at least one of the TNF blocking agents were excluded. The effects of the TNF blockers were assessed with and without a prescription history for methotrexate. We matched the diagnosis of RA or psoriasis between each drug group to a comparison (no-drug) group of patients that did not have prescriptions for any TNF blocking agents or methotrexate. The effects of each drug treatment on the risk for AD or the broader diagnosis of dementia (including AD) then were compared to the no-drug group adjusting for confounding demographic factors including age, gender, and race. **Fig 1** is a screenshot that illustrates how the Explorys Cohort Discovery platform was used to perform cohort identification for the case-control study using the Explore fast search tool.

We further investigated if other confounding factors affected the observed associations between TNF blocking agents (or methotrexate) and the reduction for risk of AD. To ensure sufficient sample size, we studied RA patients and used the broader diagnosis of dementia as the outcome measure instead of the narrower diagnosis of AD. We first investigated the effects of insurance (as a proxy of socioeconomic status). We then investigated BMI, smoking, and alcohol use as confounding environmental factors. We also investigated if the distribution of disease diagnoses and clinical findings were different between each drug group and the no-drug group. We extracted the top 50 diagnoses and the top 50 clinical findings for each drug group and the no-drug group. The common disease diagnoses (or clinical findings) between each drug group and the no-drug group are presented in S18–S25 Tables. The proportion ratios of diagnoses and clinical findings comparing the no-drug group versus a drug group are close to 1.00. Among the top 50 diagnoses, "Hypertensive disorder, systemic arterial" has the biggest difference between drug and no-drug groups. Among the top 50 clinical findings, "Non-smoker" has the biggest difference between drug and no-drug groups. Those factors are known to be associated with AD (or dementia), as suggested elsewhere [31–33]. We have considered "Non-smoker" as the major confounding environmental factor. We controlled for the diagnosis of "Hypertensive disorder, systemic arterial" and examined the associations between drugs and dementia among RA patients. Next, we investigated if the distributions of nonsteroidal anti-inflammatory drugs (NSAIDs) and steroids were different between each drug

group and the no-drug group. S26 Table shows that the proportion of patients who used NSAIDs or steroids was higher for drug groups compared to the no-drug group. Therefore, we studied the associations between drugs and dementia among RA patients by controlling for NSAIDs or steroid use status.

## Statistical analysis

The crude Odds Ratio (OR), 95% Confidence Interval (CI), and *P*-values were calculated using the Fisher exact test in univariate analysis. The Adjusted OR (AOR), 95% CI and *P*-values were calculated using the Cochran-Mantel-Haenszel (CMH) method by controlling for confounding factors such as age, gender, and race [34]. Statistical tests were conducted with significance set at *P*-value < 0.05 (two-sided).

To compute the AORs of dementia for the drug use group versus the non-drug use group, or for the inflammatory disease group versus the non-inflammatory disease group, we first calculated the OR for each of the 8 strata (tabulation of 2 age groups: 18–65, and >65 years; 2 genders: male and female; and 2 races: White and Non-White) and then calculated the weighted average across all strata using the CMH method [34].

The AOR comparing the drug use group versus the non-drug group were calculated separately in patients with a diagnosis code of RA and psoriasis. There were not enough patients (at least one cell in a two by two table have counts lower than 10 for all the strata) to test the effect of TNF blocking agents in patients with the other inflammatory diseases. We list the number of patients that were treated with each of the drugs in each of the inflammatory disease groups in S27 Table. To assess the effects of gender, age, and race on the risk associations between the therapeutic drugs and dementia, we separately calculated the crude OR of dementia among the 2 age groups, 2 gender groups, and 2 race groups. To adjust for methotrexate prescription status in psoriasis patients, we separately calculated the crude OR for a diagnosis of AD or dementia among 16 strata (tabulation of 2 age groups, 2 gender groups, 2 race groups, and 2 methotrexate groups (history of methotrexate or no methotrexate) and then calculated the weighted average across all strata using the CMH method [34].

Similarly, the AOR of co-morbid AD for each inflammatory disease group versus the control no-inflammatory disease group were calculated by weighted average across all 8 strata. To further adjust for BMI, we separately calculated the crude OR for a diagnosis of dementia among 16 strata (tabulation of 2 age groups, 2 gender groups, 2 race groups, and 2 BMI groups (BMI between 18.5 to 24.99 and BMI > = 40) and then calculated the weighted average across all strata using the CMH method. A similar method was used to adjust for smoking status (Non-smoker versus Tobacco user) and alcohol use status (Current drinker of alcohol versus Current non-drinker of alcohol).

The AOR comparing the drug use group with no-drug use group on the risk for AD were only calculated among patients > 65 years due to the limited sample size for patients <65 years that were diagnosed with both AD and a TNF-related inflammatory disease. Specifically, the AORs of AD were the weighted average of OR across all 4 strata (tabulation 2 genders: male and female; and 2 races: White and non-White) using the CMH method [34].

To investigate the effect of socioeconomic status on the association between drugs and AD (or dementia), we used patient insurance types (e.g., Medicaid, Medicare, private insurance) as a proxy for their socioeconomic status. We calculated the AOR of dementia by taking the insurance status into account. We performed this procedure for patients with age between 18–65 and patients with age > 65 years old separately, since the majority of patients with age > 65 use Medicare insurance. For patients 18–65 years old, the AOR were calculated by controlling for gender (male and female), race (White and non-White), and insurance status (private

versus no-private insurance). Patients > 65 years with low socioeconomic status receive Medicaid plus Medicare. There were not enough dementia patients for analysis who used Medicaid and who also were prescribed a TNF blocking agent. Therefore, for patients > 65 years old, the AORs were calculated among patients who did not receive Medicaid by controlling for gender and race. We adjusted BMI using a method similar to that used for inflammatory diseases. For smoking and alcohol use status, we validated the association between drugs and dementia among patients identified as non-smokers and current non-drinkers of alcohol, respectively, adjusting for age, gender, and race. We do not have enough dementia patients who are tobacco users or alcohol drinkers and who have been prescribed a TNF blocking agent. Similar to NSAIDs and steroid use, there were not enough dementia patients who have been prescribed TNF blocking agents but not prescribed NSAIDs or a steroid. Thus, we computed the AOR of dementia by comparing patients who used TNF blocking agents (or methotrexate) plus NSAIDs (or steroid) with a group of patients who received only NSAIDs (or steroids) but no TNF blocking agents and no methotrexate, adjusting for age, gender, and race.

## Results

### Structure of the study population

The IBM Watson Health Explorys Cohort Discovery platform contained EHR from 55,902,250 unique adult patients ($\geq$ 18 years of age) at the time of the study. The baseline characteristics of the study population are presented in **Table 1**. The proportion of female patients is slightly higher than the proportion of male patients (54.88% versus 44.80%), while only a small proportion of patients (176,170: 0.32%) have a missing value for gender. The proportion of White patients is higher than non-Whites (57.75% versus 42.25%). Patients between 18–65 years (70.25%) are the majority age group while only a small proportion of patients (870,670: 1.56%) have a missing value for age. 35.51% (19,851,070) of adult patients have private insurance, while 12.61% (7,048,160) are enrolled in Medicare and 6.16% (3,448,710) receive Medicaid. Since the majority of patients older than 65 years in the US transition from private insurance to Medicare, the Explorys Cohort Discovery platform contains a study population that represents the general US population and the age-dependent prevalence of AD due to the inclusion of Medicare patients. Patients with missing values for gender or age were excluded when performing statistical analysis.

In the absence of confirmatory brain imaging for amyloid or tau, or pathology post-mortem, the diagnosis of AD is a clinical diagnosis made by a physician based on an interview and cognitive assessment. Even with carefully selected patients enrolled in large, multi-center, Phase 3 clinical trials, AD is misdiagnosed based solely on clinical criteria in up to 25% of patients [35, 36]. In a database such as the Explorys Cohort Discovery platform that aggregates EHR from large, integrated health delivery systems and community-based networks, the rate of misdiagnosis of AD may be even higher. Therefore, we structured our analysis to proceed in two stages. We first calculated the AOR for a SNOMED-CT diagnosis of AD and second, expanded the analysis using the SNOMED-CT diagnosis of dementia. Within SNOMED-CT, dementia is a general diagnostic term that includes all patients with a specific diagnosis of AD. Within our study population, 338,400 (0.60%) patients were diagnosed with AD while 999,800 (1.79%) patients were diagnosed with dementia including AD. In addition to AD, the SNOMED-CT diagnosis of dementia includes subdiagnoses of presenile dementia, mild, mixed, moderate and severe dementia, dementia associated with human immunodeficiency virus infection, and frontal temporal lobe dementia, but excludes dementia associated with Huntington's disease, Parkinson's disease, Rett's syndrome and vascular dementia. The prevalence of AD in the 2018 US population is estimated at about 2% of persons older than 18 years, so

the Explorys study population of dementia including AD approximates the prevalence of AD in the US [37].

Based on prescription history, a substantial number of patients with suspected AD are likely classified within the Explorys Cohort Discovery platform with a diagnosis of dementia without the specific subdiagnosis of AD. For example, the approved uses of the cholinesterase inhibitors (donepezil, galantamine, and rivastigmine) and memantine by the US Food and Drug Administration (FDA) are for treating mild, moderate or severe dementia of the Alzheimer's type. Rivastigmine has an additional FDA approval for use in mild-to-moderate dementia associated with Parkinson's disease. Of 338,400 patients with a diagnosis code for AD, 172,750 (51%) received at least one prescription for a cholinesterase inhibitor or memantine, while an additional 157,850 patients with a diagnosis code for dementia were also treated with at least one drug approved for AD (24%). A high proportion of these patients likely represent cases of dementia due to suspected AD. Querying more broadly in the Explorys Cohort Discovery platform for any use of a cholinesterase inhibitor or memantine, 145,860 patients were prescribed at least one of the four drugs without a SNOMED-CT diagnosis of dementia or AD. This included 9,540 patients with a diagnosis code for Parkinsonism, 3,900 with a diagnosis code for degenerative disease of the central nervous system, and 3,990 patients with a diagnosis code for traumatic brain injury. Thus, Explorys captures both approved and off-label use of prescription drugs.

## Risk for Alzheimer's disease is increased by systemic inflammation

We hypothesized that systemic inflammatory diseases which in part are mediated by TNF increase the risk for AD. We therefore selected for study those inflammatory diseases for which a TNF blocking agent is an FDA approved treatment [5]. Approved indications include rheumatoid arthritis, ankylosing spondylitis, psoriasis, psoriatic arthritis, inflammatory bowel disease, ulcerative colitis, and Crohn's disease. Five TNF blocking agents have been approved by FDA, of these, sufficient patients were prescribed etanercept (44,210 patients), adalimumab (66,820 patients), and infliximab (40,290) to allow inclusion in the retrospective, case-control study. In addition, 161,560 patients received at least one prescription for methotrexate but had not been treated with a TNF blocking agent. Methotrexate is a small molecule inhibitor of immune cell activation that suppresses systemic TNF [38]. In normal clinical practice, patients are started on methotrexate and those who fail methotrexate may progress to a treatment with a TNF blocking agent. Of those who progress to the TNF blocking agent, 60% will continue on methotrexate and 40% will drop the methotrexate [38, 39].

The cumulative age distribution for patients with a diagnosis code for rheumatoid arthritis, dementia or AD is shown in **Fig 2**. Given that the Explorys Cohort Discovery platform includes at least some information on patients making up over 20% of the US population, the cumulative age distribution should reasonably represent the age-dependent prevalence of these diseases in the US. As expected, the age distribution of patients with a diagnosis of RA skews younger, while patients with dementia or AD skew older. The average treatment retention time varies among the three TNF blocking agents selected for study with etanercept having a better than 50% patient retention rate at 12 years of treatment, while the 50% retention rates for adalimumab and infliximab are about 2 and 3 years, respectively [39, 40]. To comply with HIPAA compliant de-identification of EHR, the Explorys Cohort Discovery platform does not allow patient level access to prescription data. Therefore, we compared the benefit of etanercept to adalimumab and infliximab as a surrogate for duration of exposure as a modifier of treatment benefit.

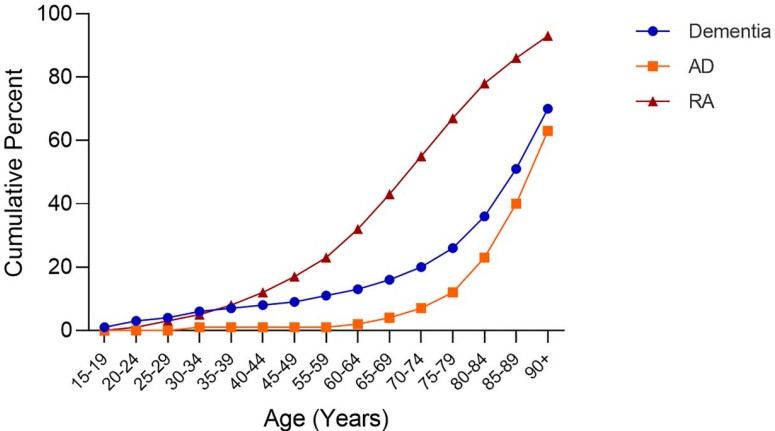

**Fig 2. Cumulative age distribution of patients with a diagnosis code for rheumatoid arthritis, dementia or Alzheimer's disease.**

Systemic inflammatory disease is a substantial risk factor for AD (**Fig 3A**). Greatest risk is associated with inflammatory bowel disease and Crohn's disease followed by rheumatoid arthritis, ulcerative colitis (as a subcategory of inflammatory bowel disease), ankylosing spondylitis and psoriasis. There is no additional risk associated with psoriatic arthritis. Inflammatory bowel disease, Crohn's disease and RA were associated with a higher risk for AD of over two-fold (AOR 2.06 (2.02–2.10)–AOR 2.46 (2.33–2.59)). After further adjusting for additional confounding environmental factors (BMI, smoking status, and alcohol use status), the positive association between RA and AD remained statistically significant, but to a lesser degree. The results are shown in **Fig 3B**.

The increased risk associated with systemic inflammation was replicated in patients with a diagnosis code for dementia in which AD is a subcategory (**Fig 4**). The risk for dementia was elevated two-fold in patients with rheumatoid arthritis, inflammatory bowel disease, ankylosing spondylitis, ulcerative colitis and Crohn's disease (AOR 2.00 (1.94–2.06) -AOR 2.69 (2.66–2.72)). Psoriasis was also associated with a higher risk for AD and dementia although to a lesser extent (AOR 1.37 (1.31–1.42) and AOR 1.49 (1.46–1.53), respectively). Thus, systemic inflammation was associated with a higher risk for AD across multiple diseases involving the joints, the gut and the skin.

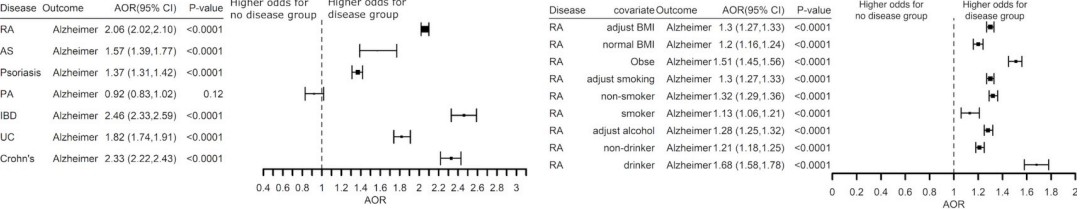

**Fig 3. a.** Adjusted Odds Ratio (AOR) for a diagnosis of Alzheimer's disease associated with a diagnosis for an inflammatory disease as compared to the non-inflammatory disease group. Abbreviations: RA -rheumatoid arthritis, AS -ankylosing spondylitis, PA -psoriatic arthritis, IBD -inflammatory bowel disease, UC -ulcerative colitis, Crohn's -Crohn's disease. **b** Adjusted Odds Ratios (AORs) for a diagnosis of Alzheimer's disease associated with a diagnosis for rheumatoid arthritis as compared to the non-inflammatory disease group, further adjusting for BMI, smoking status, or alcohol use. The added covariate is presented in column 2. We present both adjusted and stratified AOR for each covariate.

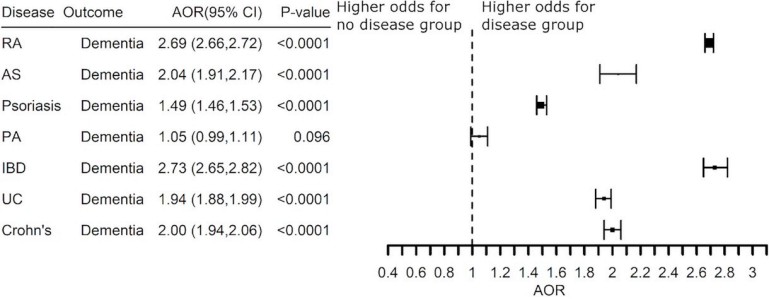

**Fig 4. Adjusted Odds Ratio (AOR) for a diagnosis code of dementia associated with a diagnosis for an inflammatory disease as compared to the non-inflammatory disease group.** Abbreviations: RA -rheumatoid arthritis, AS -ankylosing spondylitis, PA -psoriatic arthritis, IBD -inflammatory bowel disease, UC -ulcerative colitis, Crohn's -Crohn's disease.

## TNF blocking agents reduce associated risk for Alzheimer's disease in patients with systemic inflammation

If the increased risk for AD is associated with TNF, then TNF blocking agents should reduce risk for AD in patients with co-morbid inflammatory disease. Since TNF blocking agents frequently are prescribed as an add-on to methotrexate, we compared subjects who received a TNF blocking agent without methotrexate with those who received the TNF blocking agent and methotrexate. Treatment of RA with etanercept, adalimumab, or infliximab was associated with significantly lower risk for AD with etanercept and adalimumab having the greatest benefit (AOR 0.34 (0.25–0.47) and AOR 0.28 (0.19–0.39), respectively), **Fig 5**. The benefit of etanercept and adalimumab was replicated in patients with a diagnosis code of dementia (AOR 0.30 (0.26–0.35) and AOR 0.35 (0.30–0.41), respectively). Etanercept and adalimumab have similar benefit even though the treatment retention time with etanercept (50% at 12 years) is considerably longer than adalimumab (50% at 2 years) [39, 40]. Infliximab had slightly greater benefit than methotrexate, but methotrexate alone was associated with a lower risk for AD and dementia (AOR 0.52 (0.39–0.69) and AOR 0.64 (0.61–0.68), respectively), **Fig 5**. We therefore asked if the benefit of the TNF blocking agents replicated in patients with a prescription history of a TNF blocker and methotrexate compared to methotrexate alone. Etanercept and adalimumab had similar effects on the risk for co-morbid AD in patients who also had a prescription history for methotrexate (AOR 0.53 (0.42–0.67) and AOR 0.61 (0.49–0.77),

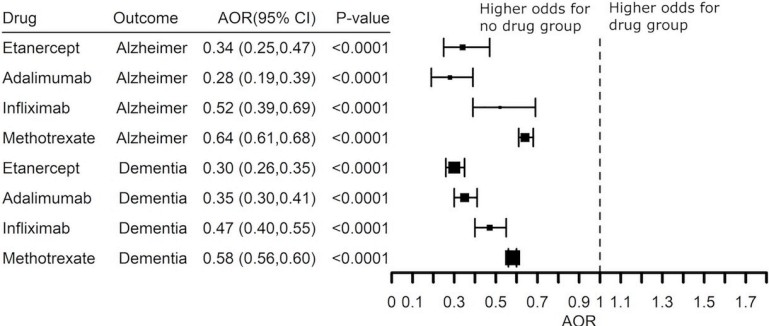

**Fig 5. Adjusted Odds Ratio (AOR) showing inverse risk associations between Alzheimer's disease or a diagnosis of dementia and prescription history of a TNF blocker (exclude certolizumab pegol and golimumab) or methotrexate compared to the no-drug group in patients with a diagnosis of rheumatoid arthritis with a prescription.** The analysis excluded patients with a prescription history of a TNF blocker and methotrexate.

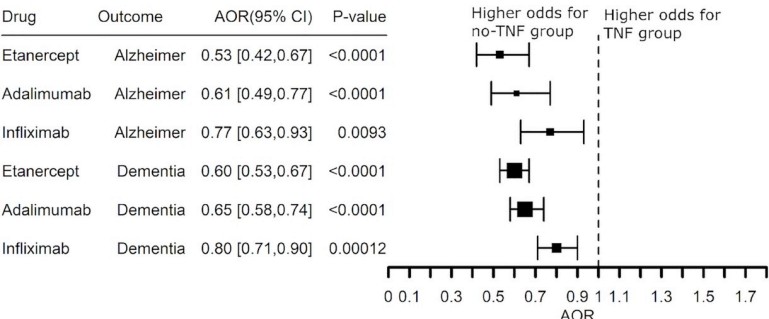

**Fig 6. Adjusted Odds Ratio (AOR) showing inverse associations between Alzheimer's disease or a diagnosis of dementia and prescription history of methotrexate and a TNF blocker compared to the methotrexate and no-TNF blocker group in patients with a diagnosis of rheumatoid arthritis.**

respectively) while the effect of infliximab was less (AOR 0.77 (0.63–0.67), **Fig 6**. S1–S6 Figs show that the significant inverse associations between drugs and dementia remain similar controlling for BMI, smoking, alcohol use, diagnosis of "Hypertensive disorder, systemic arterial", NSAIDs use or steroid use.

Lower risk for AD also was seen in psoriasis patients treated with a TNF blocking agent (**Fig 7**). Etanercept (AOR 0.47 (0.30–0.73) and adalimumab (AOR 0.41 (0.20–0.76) were associated with significantly lower risk for AD in patients with psoriasis and the benefit of adalimumab (AOR 0.54 (0.44–0.66) was replicated when expanded to patients with a diagnosis code for dementia (**Fig 7**). Infliximab and methotrexate had no benefit. A substantial number of psoriasis patients had a prescription history of a TNF blocker with methotrexate. Methotrexate alone was not associated with reduced risk for AD, nor was use of a TNF blocker associated with additional lower risk adjusting for methotrexate prescription status beyond the benefit of etanercept or adalimumab alone (AOR 0.53 (0.37–0.76) and AOR 0.50 (0.32–0.78), **Fig 8**. The Explorys Cohort Discovery platform does not permit access to patient level data, so a limitation of this analysis is that we do not know if the TNF blocker was prescribed as an add-on to methotrexate or if the two drugs were prescribed sequentially with methotrexate dropped once treatment with the TNF blocker was started.

We separately assessed the effects of gender, age, and race on the effect of TNF blocking agents on the risk of dementia among patients with rheumatoid arthritis. Data are shown for etanercept in **Fig 9**. Female and male patients had a similar benefit when treated with etanercept ($P = 0.14$), adalimumab ($P = 0.14$), or infliximab ($P = 0.71$). There was a significant difference in benefit when comparing White versus non-White patients with a prescription history for etanercept ($P = 0.022$) but not for adalimumab ($P = 0.34$) or infliximab ($P = 0.078$). Interestingly, the risk reduction was greatly increased for younger patients (18–65 years) as compared to older patients ($> 65$ years) with AOR 0.16 (0.11–0.22) in younger patients as compared to AOR 0.40 (0.34–0.46) in older patients (**Fig 9**). Greater benefit also was seen in younger patients prescribed adalimumab ($P = 0.022$) or infliximab ($P = 0.00026$) as compared to older patients.

We also explored insurance status as a potential confound for age (comparing younger patients with private insurance as compared to patients over 65 years with Medicare) and socioeconomic status (patients with Medicaid or private insurance). There was no difference between patients with private insurance as compared to Medicare, nor does adjusting insurance status influence the effect of TNF blockers on the risk for dementia (**Fig 10**). There were too few patients with Medicaid and AD (21,030, 6.2%) or Medicaid and a diagnosis of dementia (84,040, 8.4%) for analysis (**Table 1**).

Finally, we sought to understand if treatment with a TNF blocking agent was associated with decreased risk for AD compared to the general population in the absence of inflammatory disease as a risk factor and excluding a prescription history for a TNF blocker or methotrexate. Indeed, the risk for AD was lower than the general population risk in RA patients treated with adalimumab (AOR 0.62 (0.43–0.89) and in psoriasis patients treated with etanercept (AOR 0.58 (0.37–0.90) or adalimumab (AOR 0.48 (0.23–0.88), **Fig 11**. The benefit of adalimumab also replicated in psoriasis patients for decreased risk of dementia (AOR 0.82 (0.67–1.00), **Fig 11**.

## Discussion

This study shows that inflammatory diseases involving TNF are associated with increased risk for AD. Patients with rheumatoid arthritis, inflammatory bowel disease, or Crohn's disease are at highest risk for AD. For RA and psoriasis, treatment with a biologic drug that targets TNF correspondingly is associated with decreased risk for AD.

This case-control analysis provides an assessment of the additional risk for AD that can be attributed to co-morbid inflammatory disease involving TNF. Although there are genetically-defined subgroups at high risk for AD, for example, patients with highly penetrant, autosomal dominant APP, PS1 or PS2 mutations,[1] late-onset AD generally is viewed as a heterogenous disease with multiple genetic, medical and environmental risk factors [2, 41]. Our analysis suggests that co-morbid inflammatory disease defines a subgroup of Alzheimer's patients in which systemic production of TNF contributes to pathogenesis. Systemic TNF may promote neuroinflammation in the brain through receptor-mediated transcytosis [12]. Biologic drugs such as etanercept and adalimumab distribute poorly to brain, so we presume the TNF blockers act systemically to prevent TNF from reaching the brain and thereby prevent or delay the onset of AD.

Our study indicates that systemic inflammatory conditions involving TNF are a potentially treatable risk factor for AD. This is a small, but potentially treatable subgroup of people at risk for AD. The total number of patients with AD in the US is estimated to be 5.3 million in 2018 [37]. Based on our estimate of population attributable risk, 0.21 million cases (4.04%) of AD might be preventable by treating RA with a TNF blocking agent, 0.08 million cases (1.51%) of AD might be preventable by treating Crohn's disease with a TNF blocking agent, and 0.026 million cases (0.50%) of AD might be preventable by treating psoriasis with a TNF blocker or nearly 6% of cases of AD might be prevented by treatment with a TNF blocker.

Etanercept previously was evaluated for symptomatic benefit in patients with mild-to-moderate Alzheimer's disease in a small, placebo-controlled study enrolling 41 subjects [42]. End

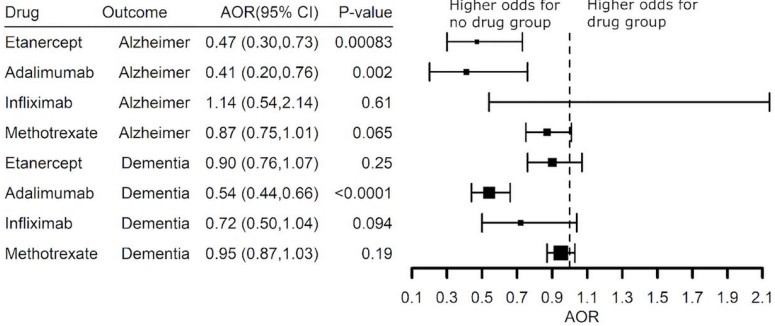

**Fig 7. Adjusted Odds Ratio (AOR) showing inverse associations between Alzheimer's disease or a diagnosis of dementia and prescription history of a TNF blocker (exclude certolizumab pegol and golimumab) or methotrexate compared to the no-drug group in patients with a diagnosis of psoriasis.**

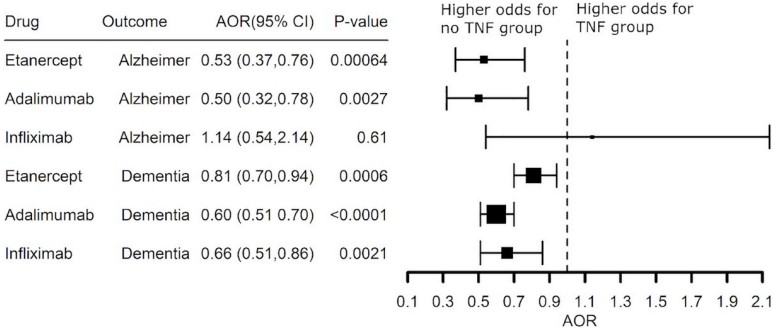

**Fig 8. Adjusted Odds Ratio (AOR) showing inverse associations between Alzheimer's disease or a diagnosis of dementia and prescription history of a TNF blocker in patients with a diagnosis of psoriasis adjusting for methotrexate prescription status.**

points in the study assessed improvement in cognitive function and activities of daily living. There was a trend for etanercept benefit at 24 weeks of treatment across multiple assessments. In contrast, our retrospective case study which used conversion to Alzheimer's disease or dementia as an outcome measure suggests the potential neuroprotective benefit of interrupting TNF signaling between the periphery and the brain. This benefit could translate to other neurodegenerative diseases as well as to psychiatric disorders. A previous study of Parkinson's disease in patients with co-morbid inflammatory bowel disease reported that TNF blocking agents were associated with decreased incidence of Parkinson's disease[43], while therapeutic benefit was reported earlier for reducing risk of major depression in a pivotal, Phase 3 clinical trial of etanercept for the treatment of psoriasis [44].

Supporting evidence for a role of TNF in AD comes from the Accelerating Medicines Partnership–AD (AMP-AD), a large-scale, precompetitive public private partnership led by the National Institute on Aging (NIA). Nominated targets with supporting evidence include multiple elements of the TNF signaling pathway such as TLR4, NFKBIA, NFKBIZ, TNFSR1A [45]. TLR4 encodes the Toll like receptor 4, a receptor in microglia, monocytes and macrophages that triggers TNF release when activated by lipopolysaccharide, or in the case of microglia, by amyloid beta peptide [41]. TLR4 and a second Toll-like receptor, TLR5, are upregulated in the

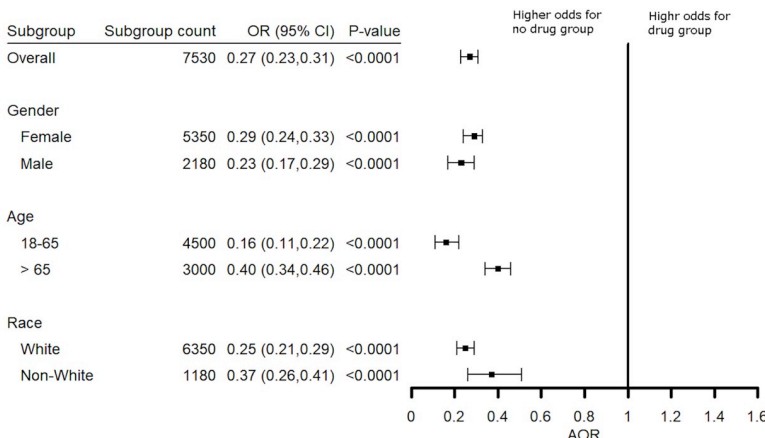

**Fig 9. Effect of age, gender and race on risk for a diagnosis of dementia in rheumatoid arthritis patients treated with etanercept.** Young patients (18–65 years) showed greater benefit than did older patients (over 65 years). The effect of gender and race was not significant.

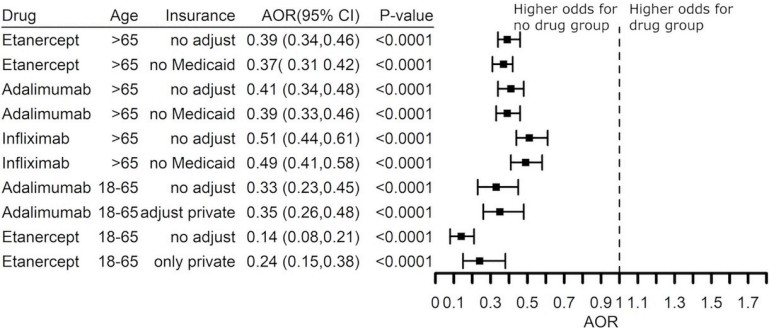

**Fig 10. Crude Odds Ratios (OR) and Adjusted Odds Ratios (AOR) based on insurance status (Medicaid versus no-Medicaid, or private versus no-private insurance) in patients with a diagnosis code for dementia and rheumatoid arthritis.**

Alzheimer's brain and soluble fragments have been shown to act as decoy receptors to ameliorate disease phenotypes in a transgenic mouse model of AD [46]. NFKBIA and NFKBIZ are components of the nuclear I kappa B protein signaling pathway that regulates the expression of TNF while TNFSR1A encodes the TNF receptor superfamily member 1A that is activated by TNF [47].

Microglia activation and gliosis formerly were considered secondary to neurodegeneration in AD, however, recent genetic studies in late-onset AD implicate microglia- and astrocyte-related pathways in disease pathogenesis.[10, 48, 49] One focus of interest is TREM2, an innate immune receptor expressed on microglia and myeloid cells [50, 51]. Rare TREM2 mutations associated with AD suggest that TREM2 deficiency contributes to AD risk [52, 53]. TREM2 is a negative regulator of the release of TNF and other inflammatory cytokines through activation of the Toll-receptor pathway [8, 51, 54]. Loss of TREM2 function in monocytes or macrophages may contribute to TNF production systemically, and indeed, be a treatable risk factor for AD much as these epidemiologic data suggest that TNF production in systemic inflammatory diseases affecting the joints, gut and skin contributes to risk for AD and that risk can be lowered by a TNF blocking agent.

TNF signaling can be targeted by small molecule therapeutics [7, 9, 55]. For example, this pathway is modulated in monocytes, macrophages and microglia by phosphodiesterase-4B (PDE4B) [56, 57], while treatment with a PDE4B allosteric inhibitor after acute brain injury reduces microglial activation, reduces production of TNF, and provides neuroprotective benefit in rats [58]. Small molecule drugs targeting the TLR4-TNF pathway may produce greater therapeutic benefit than large, biologic drugs for which brain penetration may be limited.

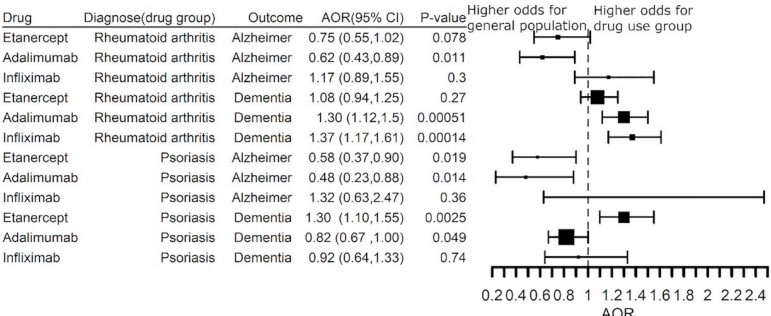

**Fig 11. Comparison of the risk for a diagnosis of Alzheimer's disease or dementia in patients with a diagnosis of rheumatoid arthritis or psoriasis with a prescription history for etanercept, adalimumab, or infliximab who were not prescribed methotrexate against the general population.**

Epidemiologic studies previously suggested that non-steroidal anti-inflammatory drugs (NSAIDs) may reduce the risk for AD [59]. In a case-control study of 49,349 US Veterans, protective effects were seen with ibuprofen, naproxen and other NSAIDs with > 5 years of use (OR 0.76 [0.68–0.85]). Use of an NSAIDs was most frequently associated with osteoarthritis (46–48% of cases and controls). The risk reduction associated with use of an NSAIDs was less than the risk reduction of AD due to use of methotrexate in patients with RA (OR 0.51 [0.49–0.54]), **Fig 3**. Multiple randomized, placebo-controlled clinical trials have failed to show benefit of treatment with an NSAIDs in prodromal or symptomatic AD [60–62]. NSAIDs are one of three categories of therapeutics used to treat rheumatoid arthritis.[5] The other categories are corticosteroids and disease-modifying antirheumatic drugs (DMARDs). NSAIDs are used for the control of pain and inflammation while corticosteroids and DMARDs are used to reduce the cellular immune response that leads to destruction of the joint. NSAIDs suppress prostaglandin synthesis by inhibiting the cyclooxygenase enzymes COX-1 and COX-2, and thereby reduce immune cell activation, but may increase TNF production [55, 63]. The lack of efficacy seen with NSAIDs in AD may be associated with the lack of a strong suppressive effect on TNF production. Low dose prednisone (10 mg/day) is used to treat rheumatoid arthritis, presumably by suppressing immune activation and TNF production, but long-term use is associated with increasingly unacceptable side effects. Treatment with low dose prednisone for 1 year failed to show benefit in patients with mild-to-moderate AD [64].

## Strengths and limitations

The strength of the current study is that the massive-scale of the EHR database allowed us to access real-world patient populations. With 64 million unique patient EHR, the IBM Watson Health Explorys Cohort Discovery platform has aggregated the medical records from approximately 20% of the US population. Such a massive database covers the full spectrum of disease diagnoses and the use of prescription drugs for both approved and off-label indications. Access to large data sets is particularly useful in the case of AD where a substantial fraction of patients in a clinical setting are either misdiagnosed as having the disease when they do not or for whom the physician is unwilling to make a specific diagnosis even though AD is suspected and an approved medication for AD is prescribed. This allowed us to test for the association of co-morbid AD with systemic inflammatory disease and to test the benefit of TNF blocking agents by first selecting patient cohorts with a specific diagnosis of AD and then by replicating that finding in a larger cohort of patients with dementia which better reflects the uncertainty of the diagnosis of AD in a community health setting.

The scale of the Watson Health Explorys Cohort Discovery platform allowed us to maintain adequate statistical power while we studied disease association and the benefit of TNF blockers across multiple inflammatory diseases. This often is difficult in traditional observational studies which frequently are limited in sample size. Compared to the private insurance database used in the previous study [13], the Explorys Cohort Discovery platform contains a study population that includes patients with private insurance and Medicare. This is particularly critical for the study of diseases that primarily affect an elderly population as the majority of patients older than 65 years in the US transition from private insurance to Medicare.

The current study has limitations. First, the quality or incompleteness of the EHR for individual patients may affect the sensitivity and specificity of the analysis and misdiagnosis will contribute to false positives and false negatives within the patient cohorts selected for study Second, the retrospective case-control analysis suggests a potential therapeutic benefit of TNF blocking agents in AD, however, the analysis does not distinguish an effect on treatment of prevention, nor does our analysis address how the severity of disease may influence benefit. A

**Table 2. Comparison of etanercept, adalimumab, and infliximab benefit across studies.**

| Study | Sample size | Outcome | TNF Blockers | OR | *P*-value |
|---|---|---|---|---|---|
| Chou 2016 [13] | 8.5 million | Alzheimer's disease | etanercept | 0.30 | 0.02 |
| The Washington Post [65] | 254,000 | Alzheimer's disease | etanercept | 0.36 | <0.0001 |
| Our study | 56 million | Alzheimer's disease | etanercept | 0.34 | <0.0001 |
| Our study | 56 million | Dementia | etanercept | 0.30 | <0.0001 |
| Chou 2016 [13] | 8.5 million | Alzheimer's disease | adalimumab | 0.65 | 0.71 |
| Our study | 56 million | Alzheimer's disease | adalimumab | 0.28 | <0.0001 |
| Our study | 56 million | Dementia | adalimumab | 0.35 | <0.0001 |
| Chou 2016 [13] | 8.5 million | Alzheimer's disease | infliximab | 0.73 | 0.68 |
| Our study | 56 million | Alzheimer's disease | infliximab | 0.64 | <0.0001 |
| Our study | 56 million | Dementia | infliximab | 0.47 | <0.0001 |

limitation of the Explorys Cohort Discovery platform is the inability to access patient level data for reasons of confidentiality. We are unable to extract age at diagnosis, duration of treatment, or years of follow up which are potential confounding factors to this analysis, nor are we able to build an exposure-response relationship for the reduction in risk due to use of a TNF blocking agent. Due the fact that we used population-level data, we relied on the odds ratio for the statistical analysis and cannot calculate an incidence rate ratio [43]. Additionally, the Explorys Cohort Discovery platform considers patients to be "taking a medication" if at least on outpatient prescription has been written for the medication, but does not record if medications were prescribed sequentially after the first drug failed, or whether two or more medications were prescribed concurrently. These potential confounds may lead our findings to underrepresent the "true findings." Without access to patient level data, we were only able to control for known confounding factors using the Cochran-Mantel-Haenszel (CMH) method [34] instead of systematically identifying and controlling for potential confounding factors using a more rigorous multivariable regression method. Even with these limitations, the results of our study are consistent with a previous study using patient-level data [13] and with a unpublished study reported in the popular press. As shown in Table 2, the therapeutic benefit of etanercept on risk for co-morbid AD that we report in patients with RA is consistent with a previous case-control analysis of 8.5 million privately insured patients [13] and with an unpublished study that reportedly analyzed claims data from 254,000 patients. Thus, our study conducted with a massive EHR database is consistent with smaller studies conducted with patient-level data collected using private insurance claims databases.

## Conclusions

Our analysis demonstrates the value of a large, population-based database aggregating EHR from nearly 56 million adult patients for the rapid interrogation of the treatment benefit of prescription drugs. Patients diagnosed with a systemic inflammatory disease are at increased risk for developing AD, while TNF blocking agents were associated with decreased risk for co-morbid AD in real-world patients diagnosed with RA or psoriasis.

## Supporting information

**S1 Fig. Adjusted Odds Ratio (AOR) showing the inverse risk association between dementia and TNF blocker (exclude certolizumab pegol and golimumab) or methotrexate compared to the no-drug group adjusting for age, gender, race, and BMI in patients with a diagnosis of rheumatoid arthritis.** The analysis excluded patients with both a TNF blocker and

methotrexate.
(DOCX)

**S2 Fig. Adjusted Odds Ratio (AOR) showing the inverse risk association between dementia and TNF blocker or methotrexate compared to the no-drug group adjusting for age, gender, and race in patients with a diagnosis of rheumatoid arthritis who are also non-smokers.**
(DOCX)

**S3 Fig. Adjusted Odds Ratio (AOR) showing the inverse risk association between dementia and TNF blocker or methotrexate compared to the no-drug group adjusting for age, gender, and race in patients with a diagnosis of rheumatoid arthritis who are also current non-drinker of alcohol.**
(DOCX)

**S4 Fig. Adjusted Odds Ratio (AOR) showing the inverse risk association between dementia and TNF blocker or methotrexate compared to the no-drug group adjusting for age, gender, race in patients with a diagnosis of rheumatoid arthritis and "hypertensive disorder, systemic arterial".** The analysis excluded patients with both a TNF blocker and methotrexate.
(DOCX)

**S5 Fig. Adjusted Odds Ratio (AOR) showing the inverse risk association between dementia and TNF blocker or methotrexate compared to the no-drug group adjusting for age, gender, and race in patients with a diagnosis of rheumatoid arthritis who also prescribed NSAIDs.**
(DOCX)

**S6 Fig. Adjusted Odds Ratio (AOR) showing the inverse risk association between dementia and TNF blocker or methotrexate compared to the no-drug group adjusting for age, gender, and race in patients with a diagnosis of rheumatoid arthritis who also prescribed steriods.**
(DOCX)

**S1 Table. Two by two tables for each of the strata comparing the odds for a diagnosis of Alzheimer's disease associated with a diagnosis for an inflammatory disease versus the non-inflammatory disease group.**
(XLSX)

**S2 Table. Two by two tables for each of the strata comparing the odds for a diagnosis of dementia associated with a diagnosis for an inflammatory disease versus the non-inflammatory disease group.**
(XLSX)

**S3 Table. Two by two tables for each of the strata comparing the odds for a diagnosis of Alzheimer's disease or dementia in patients with a diagnosis of rheumatoid arthritis with a prescription history of a TNF blocker or methotrexate versus the no-drug group.** The analysis excluded patients with a prescription history of a TNF blocker and methotrexate.
(XLSX)

**S4 Table. Two by two tables for each of the strata comparing the odds for a diagnosis of Alzheimer's disease or dementia in patients with a diagnosis of rheumatoid arthritis with a prescription history of a TNF blocker and methotrexate versus the no-TNF blocker but**

**with methotrexate patient group.**
(XLSX)

**S5 Table. Two by two tables for each of the strata comparing the odds for a diagnosis of Alzheimer's disease or dementia in patients with a diagnosis of psoriasis with a prescription history of a TNF blocker or methotrexate versus the no-drug group.** The analysis excluded patients with a prescription history of a TNF blocker and methotrexate.
(XLSX)

**S6 Table. Two by two tables for each of the strata comparing the odds for a diagnosis of Alzheimer's disease or dementia in patients with a diagnosis of psoriasis with a prescription history of a TNF blocker versus the no-drug group adjusting for methotrexate prescription status.**
(XLSX)

**S7 Table. Two by two tables investigating the effects of age, gender and race on the association between the prescription of a TNF blocker and dementia (or Alzheimer's disease) in patients with a diagnosis code for rheumatoid arthritis.**
(XLSX)

**S8 Table. Two by two tables calculating the insurance status-adjusted and crude ORs of dementia for each TNF blocker use group compared to the non-drug use group in patients with a diagnosis of rheumatoid arthritis.** The analysis excluded patients with a prescription history of a TNF blocker and methotrexate.
(XLSX)

**S9 Table. Two by two tables for each of the strata comparing the odds for a diagnosis of Alzheimer's disease or dementia in rheumatoid arthritis or psoriasis patients who have prescribed a TNF blocker but no methotrexate prescription history against the general population.**
(XLSX)

**S10 Table. Two by two tables for each of the strata comparing the odds for a diagnosis of Alzheimer's disease associated with a diagnosis for rheumatoid arthritis versus the non-inflammatory disease group further adjusting for BMI, smoking status, or alcohol use status.**
(XLSX)

**S11 Table. Two by two tables for each of the strata comparing the odds for a diagnosis of dementia with a prescription history of a TNF blocker or methotrexate versus the no-drug group among patients with a diagnosis of rheumatoid arthritis, adjusting for BMI.**
(XLSX)

**S12 Table. Two by two tables for each of the strata comparing the odds for a diagnosis of dementia with a prescription history of a TNF blocker or methotrexate versus the no-drug group among patients with a diagnosis of rheumatoid arthritis who are also non-smokers.**
(XLSX)

**S13 Table. Two by two tables for each of the strata comparing the odds for a diagnosis of dementia with a prescription history of a TNF blocker or methotrexate versus the no-drug group among patients with a diagnosis of rheumatoid arthritis who are also "current non-drinker of alcohol".**
(XLSX)

**S14 Table. Two by two tables for each of the strata comparing the odds for a diagnosis of dementia with a prescription history of a TNF blocker or methotrexate versus the no-drug group among patients with diagnoses both of rheumatoid arthritis and "Hypertensive disorder, systemic arterial".**
(XLSX)

**S15 Table. Two by two tables for each of the strata comparing the odds for a diagnosis of dementia with a prescription history of a TNF blocker (or methotrexate) plus NSAIDs versus the NAID but no TNF blockers and methotrexate group among patients with a diagnosis of rheumatoid arthritis.**
(XLSX)

**S16 Table. Two by two tables for each of the strata comparing the odds for a diagnosis of dementia with a prescription history of a TNF blocker (or methotrexate) plus steroids versus the steroids but no TNF blockers and methotrexate group among patients with a diagnosis of rheumatoid arthritis.**
(XLSX)

**S17 Table. SNOMED-CT codes of diseases investigated in the study.**
(DOCX)

**S18 Table. Top 50 common diagnoses between the etanercept group and the no-drug group.**
(DOCX)

**S19 Table. Top 50 common clinical findings between the etanercept group and the no-drug group.**
(DOCX)

**S20 Table. Top 50 common diagnoses between the adalimumab group and the no-drug group.**
(DOCX)

**S21 Table. Top 50 common clinical findings between the adalimumab group and the no-drug group.**
(DOCX)

**S22 Table. Top 50 common diagnoses between the infliximab group and the no-drug group.**
(DOCX)

**S23 Table. Top 50 common clinical findings between the infliximab group and the no-drug group.**
(DOCX)

**S24 Table. Top 50 common diagnoses between the methotrexate group and the no-drug group.**
(DOCX)

**S25 Table. Top 50 common clinical findings between the methotrexate group and the no-drug group.**
(DOCX)

**S26 Table. Counts and proportions of patients with a prescription history of NSAIDs (or steroids) in each of the drug groups.**
(DOCX)

**S27 Table. The number of patients treated with each of the drugs in each of the inflammatory disease groups.** *No-drug group represents patients who did not have prescriptions of any of the TNF blockers or methotrexate.
(DOCX)

## Acknowledgments

The authors thank Dr Susan Petanscka PhD, Dr Scott Reines MD PhD and Dr Richard Martin MD for their comments on the design of the study.

## Meeting presentation

Portions of this study were presented at the National Institute on Aging-Alzheimer's Association symposium on "Enabling Precision Medicine For Alzheimer's Disease Through Open Science", July 11-July 12, 2019, Los Angeles, California and at the meeting "Clinical Trials in Alzheimer's Disease", December 6, 2019, San Diego, California.

## Author Contributions

**Conceptualization:** Rong Xu, David C. Kaelber, Mark E. Gurney.

**Formal analysis:** Mengshi Zhou.

**Methodology:** Mengshi Zhou.

**Supervision:** Rong Xu.

**Visualization:** Mengshi Zhou.

**Writing – original draft:** Mengshi Zhou, Mark E. Gurney.

**Writing – review & editing:** Mengshi Zhou, Rong Xu, David C. Kaelber, Mark E. Gurney.

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
