## [Decision Letter · Decision Letter 0]

2 Dec 2019

PONE-D-19-26467

Tumor Necrosis Factor (TNF) Blocking Agents Reduce Risk for Alzheimer’s Disease in Patients with Rheumatoid Arthritis and Psoriasis

PLOS ONE

Dear Student Zhou,

Thank you for submitting your manuscript to PLOS ONE. I apologise for the slight delay in getting back to you whilst we were waiting for reviews to be completed. After careful consideration, we feel that your paper has merit but does not fully meet PLOS ONE’s publication criteria as it currently stands. Therefore, we invite you to submit a revised version of the manuscript that addresses the points raised during the review process. In particular would you consider the appropriateness of a multivariable regression analysis?

We would appreciate receiving your revised manuscript by Jan 16 2020 11:59PM. To enhance the reproducibility of your results, we recommend that if applicable you deposit your laboratory protocols in protocols.io, where a protocol can be assigned its own identifier (DOI) such that it can be cited independently in the future. For instructions see: http://journals.plos.org/plosone/s/submission-guidelines#loc-laboratory-protocols

We look forward to receiving your revised manuscript.

Kind regards,

Antony Bayer

Academic Editor

PLOS ONE

Journal Requirements:

1. Please note that according to our submission guidelines (http://journals.plos.org/plosone/s/submission-guidelines), outmoded terms and potentially stigmatizing labels should be changed to more current, acceptable terminology. For example: “Caucasian” should be changed to “white” or “of [Western] European descent” (as appropriate).

2. Thank you for including your competing interests statement; "The authors have declared that no competing interests exist."

We note that one or more of the authors are employed by a commercial company:

 Tetra Therapeutics

Reviewers' comments:

Reviewer's Responses to Questions

**Comments to the Author**

1. Is the manuscript technically sound, and do the data support the conclusions?

Reviewer #1: Yes

Reviewer #2: Yes

Reviewer #3: Yes

2. Has the statistical analysis been performed appropriately and rigorously? 

Reviewer #1: Yes

Reviewer #2: Yes

Reviewer #3: Yes

3. Have the authors made all data underlying the findings in their manuscript fully available?

Reviewer #1: Yes

Reviewer #2: No

Reviewer #3: Yes

4. Is the manuscript presented in an intelligible fashion and written in standard English?

Reviewer #1: Yes

Reviewer #2: Yes

Reviewer #3: Yes

5. Review Comments to the Author

Reviewer #1: PONE-D-19-26467

Review

This is an interesting study using big data.

Across the manuscript authors report “increased the risk”. The correct description will be “exposure (disease +/- drug) is associated with higher risk for AD”

Authors have selected a limited number of confounders. Confounding bias is one of the most limitations for an observational study like this. Although authors acknowledge this problem in the discussion I miss a deeper external validation for factors used in other studies to find association between inflammatory diseases and AD.

The authors also acknowledge that the “Cochran-Mantel-Haenszel (CMH) method instead of systematically identifying and controlling for potential confounding factors using a more rigorous multivariable regression method”. My mind concern is that authors are losing statistical power using CMH method, e.g., page 16 lines 350-351 “There were too few patients with Medicaid and Alzheimer’s disease (21,030, 6.2%) or Medicaid and a diagnosis of dementia (84,040, 8.4%) for analysis” This would not be true for a logistic regression analysis. I encourage the authors to repeat the analysis using a multivariable regression approach.

Abstract: the abstract does not say what analysis what used.

Methods:

• could the authors list SNOMED-CT codes in Appendix in order to allow reproducibility of their work

• Could the authors report how was performed the matching, please? Page 7 lines 140-144

• Insurance as a proxy for socioeconomic status seems to be of difficult interpretation for patients older than 65. Page 8 line 179.

Results

• Could the authors show in a table differences of patients with missing values excluded for analysis in terms of exposures and outcomes, please? And report how missing values were addressed in methods.

• Could the authors move to the discussion page 14 lines 317 to 320 and describe how the current/routine practice is, please?

Reviewer #2: Overall, this is an important and impressive study, leveraging an extremely powerful data source that covers a substantial portion of the US population electronic health data. The major findings are convincing and conclusions mostly well justified. However, there are some limitations including a fairly simplistic analysis. In general, the study would be strengthened by some additional data on comparisons of treated vs no treatment.

Specific comments and questions are raised below:

The AOR are calculated comparing the reduction in risk for AD/dementia with a given diagnosis (eg, RA, Figure 5) compared to the no-drug group. There may be a selection bias in the severity of disease, insurance, and/or many other factors that contribute to whether or not treatment is administered. Some effort to control these confounders should be attempted, eg., at least using other potential diagnoses and medications and propensity scores. At least providing a Table with comparisons of treated vs untreated would be very helpful.

Similarly, it is not clear if the “no-drug” group included patients with inflammatory conditions treated with other medications such as NSAIDS, or steroids. As pointed out in the discussion, “NSAIDs suppress prostaglandin synthesis by inhibiting the cyclooxygenase enzymes COX-1 and COX-2, and thereby reduce immune cell activation, but may increase TNF production [47, 55]”. Given this possibility, NSAID use may be a confound that should be addressed.

The manuscript states: “Systemic inflammatory disease is a substantial risk factor for Alzheimer’s disease (Figure 3). Greatest risk is associated with inflammatory bowel disease and Crohn’s disease followed by rheumatoid arthritis, ulcerative colitis (as a subcategory of inflammatory bowel disease), ankylosing spondylitis and psoriasis.” How many of those with each diagnosis were treated? This information may be buried in table 1 but it would help if there were explicit numbers provided in the figure or table. Were these numbers that were used to derive the OR based only on the untreated numbers of patients (eg. RA) or all patients with each diagnosis regardless of treatment condition?

“Indeed, the risk for Alzheimer’s disease was reduced below the general population risk in rheumatoid arthritis patients treated with adalimumab (AOR 0.62 (0.43 – 0.89) and in psoriasis patients treated with etanercept (AOR 0.58 (0.37 – 0.90) or adalimumab (AOR 0.48 (0.23 – 0.88), Figure 11.” However, the same figure shows that for RA, the adalimumab or infliximab treatments still yielded higher adjusted odds ratios (AOR >1) compared to the general population. This should be discussed, as not all TNF blocking agents reduce risk for Alzheimer’s disease below the risk in the general population in the absence of inflammatory disease.

Comparison of AD risk vs another CNS disorder that might not be expected to involve systemic inflammation may be a useful control. For example, consider stroke or migraine. These or other “control” comparisons would bolster confidence in the findings from this database.

I could not find the time windows that constrained the analyses. How long were records available – in years? That is, in calculating the OR for RA, how many years minimum and maximum were allowed prior to outcome of Alzheimer? Ditto for other diseases.

The analyses do not seem to adjust for multiple comparisons; given this large sample size it is critical to do so.

How about other dementia subtypes such as DLB and FTD? Dx are also usually stratified for EOAD (presenile dementia).

Medication use was defined as “taking a medication” if at least one outpatient prescription was identified. To obviate one-time use, it would be helpful to see if the effects are altered by defining as at least 2 prescriptions to help insure adequate drug exposure.

The discussion addresses the few previous clinical trials of TNF antagonist therapies in symptomatic AD. These trials all seem to be based on clinical benefit as outcomes, but the retrospective study in this paper is examining AD/dementia as an outcome. It should be pointed out that these findings suggest the potential neuroprotective effects of anti-inflammatory treatments involving TNF signaling, rather than clinical symptomatic benefits.

Reviewer #3: This is a very well done manuscript on an important topic. The writing is clear and not ambiguous. The conclusions support the concept of inflammation in Alzheimer's disease, which is a basic but little emphasized topic.

6. PLOS authors have the option to publish the peer review history of their article (what does this mean?). If published, this will include your full peer review and any attached files.

Reviewer #1: No

Reviewer #2: No

Reviewer #3: No

---

## [Author Response · Author response to Decision Letter 0]

10 Jan 2020

We greatly appreciate the reviewers’ constructive and insightful comments and suggestions, which have helped us significantly improve the manuscript. The response to reviewers have been included in the file "Response to Reviewers".

---

## [Decision Letter · Decision Letter 1]

18 Feb 2020

Tumor Necrosis Factor (TNF)Blocking Agents are Associated with Lower Risk for Alzheimer’s Disease in Patients with Rheumatoid Arthritis and Psoriasis

PONE-D-19-26467R1

Dear Dr. Zhou,

Thank you for your detailed attention to the reviewer comments on your original submission. We are pleased to inform you that your revised manuscript has been judged scientifically suitable for publication and will be formally accepted for publication once it complies with all outstanding technical requirements.

With kind regards,

Antony Bayer

Academic Editor

PLOS ONE

Additional Editor Comments (optional):

Reviewers' comments:

Reviewer's Responses to Questions

**Comments to the Author**

1. If the authors have adequately addressed your comments raised in a previous round of review and you feel that this manuscript is now acceptable for publication, you may indicate that here to bypass the “Comments to the Author” section, enter your conflict of interest statement in the “Confidential to Editor” section, and submit your "Accept" recommendation.

Reviewer #2: All comments have been addressed

2. Is the manuscript technically sound, and do the data support the conclusions?

Reviewer #2: Yes

3. Has the statistical analysis been performed appropriately and rigorously? 

Reviewer #2: Yes

4. Have the authors made all data underlying the findings in their manuscript fully available?

Reviewer #2: Yes

5. Is the manuscript presented in an intelligible fashion and written in standard English?

Reviewer #2: Yes

6. Review Comments to the Author

Reviewer #2: Not applicable. The authors have adequately revised the manuscript, addressing the issues raised in the previous reviews.

7. PLOS authors have the option to publish the peer review history of their article (what does this mean?). If published, this will include your full peer review and any attached files.

Reviewer #2: No

---

## [Editor Report · Acceptance letter]

9 Mar 2020

PONE-D-19-26467R1 

Tumor Necrosis Factor (TNF) Blocking Agents are Associated with Lower Risk for Alzheimer’s Disease in Patients with Rheumatoid Arthritis and Psoriasis 

Dear Dr. Zhou:

I am pleased to inform you that your manuscript has been deemed suitable for publication in PLOS ONE. Congratulations! Your manuscript is now with our production department. 

With kind regards,

on behalf of

Professor Antony Bayer 

Academic Editor

PLOS ONE